# Advances in Biosynthesis of Natural Products from Marine Microorganisms

**DOI:** 10.3390/microorganisms9122551

**Published:** 2021-12-10

**Authors:** Quan Zhou, Kinya Hotta, Yaming Deng, Rui Yuan, Shu Quan, Xi Chen

**Affiliations:** 1Key Laboratory of Synthetic and Natural Functional Molecule of the Ministry of Education, College of Chemistry and Materials Science, Northwest University, Xi’an 710127, China; mrzhouquan01@163.com (Q.Z.); 2021612300@hust.edu.cn (Y.D.); 18392031422@163.com (R.Y.); 2Genetech Biotechnology Pte Ltd., 8 Temasek Boulevard #34-03, Singapore 038988, Singapore; kinyah@gmail.com; 3State Key Laboratory of Bioreactor Engineering, East China University of Science and Technology, Shanghai Collaborative Innovation Center for Biomanufacturing (SCICB), 130 Meilong Rd., Shanghai 200237, China; shuquan@ecust.edu.cn

**Keywords:** marine natural products, genomic screening, heterologous expression, combinatorial biosynthesis

## Abstract

Natural products play an important role in drug development, among which marine natural products are an underexplored resource. This review summarizes recent developments in marine natural product research, with an emphasis on compound discovery and production methods. Traditionally, novel compounds with useful biological activities have been identified through the chromatographic separation of crude extracts. New genome sequencing and bioinformatics technologies have enabled the identification of natural product biosynthetic gene clusters in marine microbes that are difficult to culture. Subsequently, heterologous expression and combinatorial biosynthesis have been used to produce natural products and their analogs. This review examines recent examples of such new strategies and technologies for the development of marine natural products.

## 1. Introduction

Natural products (NPs) are secondary metabolites produced by living organisms. These compounds are often used as drugs, although their natural functions are usually not known. Since the discovery of penicillin in 1928, numerous natural products have been extracted from microorganisms, some of which have become successful clinical drugs [1]. One report summarized that 50% of the new drugs developed from 1981 to 2019 originated from nature (including NP derivatives, synthetic compounds containing NP functional groups and natural product mimics) [2]. Moreover, the complex structures and diversity of natural products have inspired many synthetic chemists to produce these molecules in the laboratory, including widely used β-lactam antibiotics (penicillin), anticancer drugs (paclitaxel), immunosuppressants (cyclosporin), and cholesterol lowering agents (lovastatin) [3].

Along with the development of deep-sea exploration technology, spectroscopy, X-ray crystallography, and various separation techniques, marine natural products (MNPs), especially microbial secondary metabolites, have attracted wide attention [4]. The extreme environment in the sea, characterized by various conditions, such as high or low temperatures, high pressure, low pH, and high salt concentrations [5], is thought to endow marine secondary metabolites with novel chemical structures and unique physiological properties because of the major driving forces for the evolution of regulated biosynthetic traits [6]. Since the first report of bioactive MNP spongothymidine in 1950, more than 30,000 MNPs have been discovered [7], which still represent just a small portion of the total number of MNPs that are believed to exist in nature. In recent years, the number of known MNPs has increased significantly. For example, in 2018 alone, more than 1500 new MNPs were reported, which is an increase of 4% compared to 2017 [8]. Many MNPs with antitumor activities, such as plitidepsin, gemcitabine, glembatumumab, pseudopterosins, PM1004, vedotin, and elisidepsin, have entered clinical trials, and some have gained approval by regulatory agencies [9]. So far, there are eight commercial drugs that originated from MNPs, including ω-conotoxin MVIIA (commercial name Prialt) which is the first MNP drug approved by the FDA to treat spinal cord injuries [10]. Moreover, there are 22 MNP-based drug candidates undergoing clinic trials [11]. Compared with the hit rate of 1/5000 to 1/10,000 of non-marine-derived NPs as drugs, the hit rate for MNPs is higher at 1/3500 [11]. Obviously, MNPs are an important source for drugs [12,13,14]. However, challenges remain in fully realizing the therapeutic potential of MNPs. Unlike terrestrial sample collection, the difficulty in reaching the marine environment, especially in the deep sea, poses problems for sample collection. The culturing of marine microorganisms is also difficult. Although there are more than one million species of marine microorganisms, less than 1% have been cultured successfully in the laboratory. Moreover, it is very difficult to obtain sufficient quantities of the secondary metabolite of interest from natural producers, which in some cases has caused erroneous determinations of the chemical structures, formulas, and biological activities of the compounds. Fortunately, modern bioinformatics and genetic tools have enabled researchers to identify biosynthetic gene clusters in the genome of microorganisms [15], which provide a new route to discover, produce, and modify MNPs. Nowadays, we no longer rely on traditional separation and purification techniques to obtain a limited quantity of NPs. Instead, we are able to use biotechnology to produce both natural and unnatural natural products.

This review covers several key aspects of MNP engineering regarding activity selection, biosynthetic gene cluster mining, heterologous biosynthesis, and combinatorial bio-synthesis. Furthermore, this review discusses several methods for MNP development that can serve as references for new MNP researches. There are a number of earlier reviews that have covered studies on MNPs. For instance, a review from 2011 described in depth the use of marine bacteria and fungi as crucial sources of MNPs and the methods and approaches available for exploring those marine microorganisms, including genomic and metagenomic techniques that are geared toward sustainably exploiting their potential for bio-mining new MNPs [16]. Other reviews discussed the lack of research efforts in the discovery of MNPs and the characterization of underrepresented marine microorganisms, such as marine fungi of the genera *Talaromyces* and *Bartalinia*, which could hold a hidden wealth of diverse MNPs with valuable bioactivities [17,18]. A recent review illustrated the development of marine Proteobacteria not only as sources of new BGCs and MNPs but also as a novel cell factory to produce MNPs [19]. Another review from 2021 also highlighted the biosynthetic studies of MNPs from different compound classes and detailed analyses of their BGCs and biosynthetic pathways [6]. To complement these reviews, our review aims to cover a broader range of marine microorganisms and focuses on practical strategies and technologies for solving the problems that may hamper research into MNPs.

## 2. The Discovery of MNPs

The ocean occupies most of the earth’s surface and close to 87% of the biosphere [20], representing a huge reservoir of biodiversity. Marine organisms produce numerous compounds with unique chemical structures and bioactivities [21]. From 1985 to 2012, 15,000 natural products have been extracted from marine organisms, 4196 of which showed bioactivities [22,23]. More than 30,000 publications have been deposited in the database of marine natural products research (http://pubs.rsc.org/marinlit, accessed on 18 November 2021), and 45 patents have been granted for therapeutic applications of MNPs between 2015 and 2018 [24]. In recent years, most MNP discoveries have come from marine microorganisms. In particular, MNPs from fungi have increased significantly, and they are expected to surpass sponges as the most abundant source of MNPs by 2024 [8].

## 3. Genome-Independent MNP Discovery

Phenotypic screening is a classical method for identifying new natural products. Phenotypic screening utilizes fermentation broths or extracts of the strains examined for activity tests. The simplest methods include cell growth inhibition or apoptosis assays. The process generally involves several steps. First, a microbe of interest is either collected from the field or curated from a source. The microbe is cultured in a suitable culture medium, and the culture broth is extracted with solvents, often involving solvent optimization to extract fractions of medium to high polarity. Then, the fractions are purified further, typically by simple column chromatography. Next, pure compounds for a certain activity are obtained by different chromatography methods, such as high-performance liquid chromatography (HPLC) coupled with suitable activity assays [22]. Abdelkarem et al. (2019) analyzed the effects of the methanol extract of Egyptian soft coral *Heteroxenia fuscescens* on a human breast adenocarcinoma (MCF-7) cell line using the MTT assay. Their analysis identified one mono-alkylglycerol and five steroids (Figure 1) and discovered that compounds **2**, 11α-acetoxy-gorgost- 3β,5α,6β-triol and **4**, (23R) methylergosta-20-ene-3β,5α,6β,17α-tetrol displayed moderate cytotoxic activities with the IC_50_ values of 33.2 and 25.1 µM, respectively [25]. Similarly, de Vera et al. cultured 33 microalgae strains and prepared the extracts to screen for antimicrobial, anti-proliferative, and apoptotic potentials [26]. Through the study, they were able to identify several hit compounds for future drug developments.

With the fast development of nuclear magnetic resonance (NMR) spectroscopy and mass spectrometry (MS), the effectiveness of phenotypic screening has improved. The combination of effect-directed analysis (EDA) chromatography with classical chemical and biochemical methods has been widely used for drug discovery, especially for target identification of bioactive molecules [27,28,29]. In general, the three-dimensional structures of MNPs are closely related to their bioactivities. For example, different chirality in the compounds may give rise to huge differences in their biological activities. The stereochemistry of a chiral center can be determined by NMR experiments, such as nuclear Overhauser effect spectroscopy (NOESY) and rotating frame Overhauser effect spectroscopy (ROESY). Similarly, X-ray crystallography can be used to determine the absolute configuration of a compound. Combining chromatographic separation and biochemical activity analysis, factions with bioactivities can be efficiently identified from extracts of microbial fermentation [30]. As an example, Hohmann et al. discovered a new antibiotic caboxamycin through HPLC-diode array screening of the extracts of the marine strain *Streptomyces* sp. NTK 937, isolated from deep-sea sediment [31]. The structure of this compound was determined by mass spectrometry and NMR. The anti-microbial, anti-biofilm formation and cytotoxic activities were also elucidated through bioassays. In another example, Kim et al. reported isolation of α- and γ-pyrones named nocapyrones from the culture extract of a *Nocardiopsis* strain collected from marine sediment [32]. Again, NMR and mass spectrometry were applied to determine the absolute configurations of these pyrone-bearing compounds with medium branched aliphatic side chains. Biological activity assays revealed that nocapyrones possessed the ability to induce preadipocyte differentiation and the production of anti-diabetic adiponectin ex vivo.

A huge number of bioactive compounds can be screened in parallel using small amounts of a sample. There are some examples of high-performance thin-layer chromatography (HPTLC) being applied to extracts to screen, detect, and quantify their antioxidant and inhibitory activities against α-amylase [27], as well as acetylcholine esterase (AChE) inhibitory activity [29]. Recently HPTLC was applied in combination with microchemical (using reagents, such as DPPH free radical, p-anisaldehyde, and Fast Blue B) as well as biochemical (e.g., enzymatic activities of AChE and α-amylase) derivatizations to discover many compounds with bioactivities from marine algae extracts [30,33,34]. These compounds can serve as leads in developing novel therapeutics for the treatment or prevention of diabetes, neurodegenerative diseases, and cancer.

Novel natural products can be obtained in new locations on earth. Through Global Natural Products Social Molecular Networking (GNPS), geographical hotspots for chemical diversity have been revealed [35]. This information can lead to the discovery of natural products which have not been characterized before. Leber et al. have developed a flexible analysis method which is called Objective Relational Comparative Analysis (ORCA) [36]. This method was applied together with GNPS, as well as LC–MS analysis, to chemical extracts from different geographical resources, including crude extracts of field-collected samples of the tropical marine cyanobacterium *Moorena bouillonii* from various sites in the Indo–Pacific region. Through this exercise, approximately ten MNPs with novel chemical structures were discovered, which can be used to screen for drug leads. Nature is a wise designer, from which unique MNPs can be discovered.

## 4. Genome-Dependent MNPs Discovery

Only a small fraction of marine microbes can be cultivated under laboratory conditions. Genomic and metagenomic analyses have revealed that the symbiosis between hosts and co-evolved microbiomes, whose members are typically unculturable on their own in the environment the host inhabits, is one of the major driving forces in the generation of various MNPs [37]. Through genomic sequencing, accompanied by bioinformatics, limitations imposed by the unculturable nature of the microbes can be overcome, and new natural product biosynthesis gene clusters (BGCs) can be discovered. Genome mining relies entirely on computational and bioinformatics tools. Nowadays, huge amounts of DNA sequence data are available from public databases. Since natural products with the same backbone are produced by similar modules, when a new gene is discovered, its sequence can be compared with known sequences in the database to predict the structure of the compound it synthesizes and prevent rediscovery. Genome mining can also be used to gauge the potential functions of the resulting metabolites of BGCs [38]. With the guidance of genomics information, the chance of discovering secondary metabolites with biological activities can be increased [39]. So far, a lot of databases have been developed for genome mining. For example, the “Antibiotics and Secondary Metabolite Analysis Shell—antiSMASH” (https://antismash.secondarymetabolites.org, accessed on 18 November 2021) is the most widely used tool to identify and analyze BGCs in the DNA sequence of bacteria, fungi, and plants [40]. AntiSMASH was launched in 2011 and has been further expanded and improved. This database can make predictions from the DNA sequences entered by users to provide detailed information about identified gene clusters. For example, the output can provide such detailed information as the stereochemistry of the amino acids for the building blocks of the predicted compound [41]. Version 5.0 of antiSMASH was released in April, 2019 [40], with updates of the predictions for β-lactones, acyl-amino acids, fungal RiPPs (ribosomally synthesized and post-translationally modified peptides), Ras-RiPPs (radical S-adenosylmethionine RiPPs) c-nucleosides, PBDEs (polybrominated diphenyl ethers), lipolanthines, and PPY-like pyrones. Its HTML output visualization capability has also been enhanced. AntiSMASH version 6.0 was released just recently in May 2021, with an increase of the number of supported cluster types from 58 to 71, more explicit detection and display of the modular structure of multi-modular megaenzymes, improved annotation for enzymes in RiPP clusters, the addition of a new BGC comparison algorithm called ClusterCompare, and the ability to crosstalk with other prediction tools [42]. PRISM (PRediction Informatics for Secondary Metabolomes, http://magarveylab.ca/prism, accessed on 18 November 2021) is another computational NP chemical structure prediction algorithm [43]. PRISM performs chemical graph-based secondary metabolite structure prediction to allow manipulation of predicted structures at the level of individual atoms for the prediction of chemical structures. The algorithm can predict for not just nonribosomal peptides (NRPs) and type I and type II polyketides (PKs) produced by modular biosynthetic enzymes but also products of non-modular biosynthetic pathways, such as alkaloids, aminoglycosides, and nucleosides, from microbial genomic sequences. RODEO (Rapid ORF Description and Evaluation Online, http://www.ripp.rodeo/index.html, accessed on 18 November 2021) is another bioinformatics tool that combines hidden-Markov-model-based analysis, heuristic scoring, and machine learning to specialize in identifying RiPP-generating BGCs and predicting the structures of the resulting products [44]. RODEO is particularly suited for identifying lasso peptides, class I lanthipeptides, sactipeptides, and thiopeptides. The Paired Omics Data Platform (PoDP, https://pairedomicsdata.bioinformatics.nl, accessed on 18 November 2021) is a new tool for paired genomic and metabolomic data mining and for establishing large-scale genome–metabolome associations which facilitates the elucidation of the structures of metabolites and the biosynthetic genes encoded in microbial genomes [45]. Specifically for antibiotics genome mining, Antibiotic Resistant Target Seeker (ARTS, http://arts.ziemertlab.com, accessed on 18 November 2021) has been developed and has accelerated the discovery of new antibiotics [46]. ARTS specializes in detecting and analyzing resistance factors associated with BGCs that can provide insight into whether or not BGCs encode antibiotics.

Genome mining combines molecular biological techniques, DNA sequencing, and bioinformatics in order to obtain information on the BGCs of an organism and predict the physical and chemical properties of the products arising from BGCs, thus validating the potential of BGCs for synthetic biology applications. This method is not influenced by the content of the products present in the organisms or expression of the BGCs, making it a highly convenient approach for expanding the scope of research on the biosynthesis of MNPs. Genetic and mass spectrometric analyses can also provide a solid basis for natural product discovery and identification. At the same time, high throughput genome screening and biochemical pathway prediction can yield big data on microbes [47]. Recently, Wang et al. analyzed 2699 genomes from bacteria, archaea, fungi, and protists [48] using two independent secondary metabolite biosynthesis enzyme tools: 2metDB [49] and antiSMASH [40]. By comparing the gene clusters, 3339 NRPs and PKs were predicted, 90% of which were unknown. In order to rapidly evaluate the potential of marine prokaryotes to produce NRPs and PKs, Amoutzias et al. performed genome mining on approximately 2700 complete prokaryotic genomes. The software HMMER [50] was utilized to analyze the hidden domains of NRP synthetases (NRPSs) and PK synthases (PKSs), which discovered nearly 300 NRPS and PKS genes [51]. Zhang et al. used a tool based on metabolomics called HCAPCA (hierarchical cluster analysis principal components analysis) [52,53,54] to classify 1482 actinobacteria from marine invertebrates collected in the Florida Keys between 2012 and 2016 and screened the metabolites of those microbes for antibiotic activities [55]. Based on titers, MS, and NMR analysis, they discovered a promising antifungal drug turbinmicin **7** (Figure 2) from *Micromonospora* sp., a member of a sea squirt microbiome. The minimum inhibitory concentration (MIC) of **7** against the panresistant *Candida auris* strain B11211 was 0.25 µg/mL, which was eight-fold lower than that of the clinical drug amphotericin B. In a different study, Xu et al. reported on targeted genome mining to identify the biosynthetic pathways for purine nucleoside antibiotics [56]. They targeted the enzyme PenB (short-chain dehydrogenase) and PenC (SAICAR synthetase) from the pentostatin (PTN) pathway to search the NCBI database by BLASTP. They discovered that both *Streptomyces citricolor* NBRC 13005 and *Micromonospora haikouensis* DSM 45626 carried BGCs that could support biosynthesis of PTN-related antibiotics. The functions of some of the enzymes encoded by the BGCs were determined, and the previously poorly characterized biosynthetic pathways for aristeromycin and coformycin were proposed. This is an example of activity screening at the genetic level.

Due to the special conditions of the environment which marine microorganisms typically inhabit, such as extreme temperature, pressure, pH, and salt concentration [5], it is inherently difficult to obtain marine microorganisms and the NPs they produce. Moreover, rediscovery of known compounds is a common problem in NP discovery [57]. Genome sequence analysis can exclude the BGCs of MNPs that have been discovered previously. Combined with high-throughput screening and prediction for new structures, directions for the discovery of novel NPs and their analogs and detection of MNPs with interesting biological activities can be facilitated greatly [58]. A major limitation of genome mining is that it can only recognize known BGCs [59]. Furthermore, this method is unable to predict all biological activities of NPs, especially those without a precedent. However, the genome mining technique is rapidly improving and has huge potential in MNP discovery. Accompanied by the development of related biological and bioinformatics tools, genome mining can efficiently increase the success rate of MNP discovery, thus promoting the development of marine drugs. One thing that needs to be noted is that prediction of BGCs based on the genome mining approach and chemical structure identification should complement each other. Identification of BGCs can supply a large number of new structures, and chemical analysis of the structures can further enrich the biological and chemical information in the database. Such a cycle of knowledge accumulation and expansion will help improve the database as a reference for future genome mining and structure prediction. As genome sequencing technology improves and its costs decrease, more than a million predicted BGCs are already available, and the number is expected to continue growing [60]. The availability of such a wealth of information, accompanied by increasing knowledge of biological activities of different NPs, could begin to allow scientists to selectively screen NPs for desired functions. As a whole, advancement in data mining has accelerated the discovery of MNPs and prevented rediscovery.

## 5. Heterologous Biosynthesis

MNPs with novel skeletons and unique activities are an important source of drug leads. However, due to the difficulty of obtaining marine samples reproducibly and the extremely low content of active constituents in these samples, it is a formidable task to obtain a sufficiently large number of MNPs to allow a thorough evaluation of their biological activities and medicinal properties, even when a sample is found to be highly active during an initial screen. Furthermore, many MNPs have a complex structure, making them difficult to synthesize, which prevents subsequent studies, such as drug validation and optimization of the structures. These are some of the critical bottlenecks in marine drug research. In recent years, scientists have successfully curated a large volume of genome sequences from the ocean and discovered many novel BGCs. Such discoveries provide a great potential for the identification of novel NPs and have renewed excitement in isolating interesting NPs from complex marine resources. However, cryptic culture conditions of marine microorganisms limit their studies and applications [61,62]. Nevertheless, deep amplicon sequencing analyses suggest the substantial biosynthetic potential of uncultivable bacteria [63]. Since the original hosts cannot be utilized directly, heterologous hosts which are easier to culture in the laboratory can be used to synthesize MNPs. This approach can bypass the problems associated with laboratory cultivation and management of original hosts and allow the production of compounds from uncultivable or symbiotic organisms. With the fast development of synthetic biology and gene engineering, many methods have been developed to deal with BGCs that do not express well under the standard conditions [61,62]. Figure 3 summarizes the general strategy of heterologous biosynthesis. First, DNA is extracted from the original host. Then the genome sequences are obtained. Next, BGCs are determined with the aid of bioinformatics tools. Finally, BGCs can be cloned and assembled into expression plasmids using appropriate methods, such as TAR (transformation-associated recombination) [64] or CATCH (Cas9-assisted targeting of chromosome segments) [65,66]. Thus, prepared recombinant plasmids are then transformed to a heterologous host for expression.

Selecting a suitable host and an appropriate vector is crucial for successful heterologous expression [67]. For the expression of BGCs from bacteria, such as *Streptomyces* and other actinomycetes, laboratory *Streptomyces* strains, such as *S. coelicolor*, *S. lividans*, and *S. albus*, are favorable because of compatible codon and promoter usage [68]. Recently, microalgae have been applied as heterologous hosts to produce proteins at high expression levels through sequence optimization [69] and using secretory signal peptides [70]. Below, we will briefly discuss two successful applications of heterologous biosynthesis in obtaining MNPs.

Diazaquinomycins (DAQs) are NPs with anti-tubercular activity [72]. Initially isolated as a weak antibiotic from a soil dwelling *Streptomyces* sp. OM-704 [73], a family of DAQ derivatives has been found in another soil *Streptomyces* strain GW48/1497 [74], as well as a marine *Streptomyces* strain F001 [75] and *Micromonospora* sp. B006 from freshwater [76]. The anti-tubercular activity of DAQs was identified more recently, but the low solubility of DAQs limited their application [72]. Understanding DAQ biosynthetic pathways would be helpful in introducing new side chains to the core structures of DAQs to improve the solubility and expand the scope of applications. Here, the MAUVE alignment [77] was applied to the genome sequence of the marine *Streptomyces* sp. strain F001 against that of the freshwater *Micromonospora* sp. strain B006 to find a shared gene cluster between the two DAQ-producing strains. The search identified a 19 kb gene cluster in the *Micromonospora* sp. genome [76]. Furthermore, the CRISPR–Cas9 system was used to delete the genes in the cluster to identify the function of this BGC and draft the proposed DAQ biosynthetic pathway (Figure 4A, bottom row). Subsequently, the corresponding 23.5 kb BGC from *Streptomyces* sp. F001 (Figure 4A, top row) was successfully amplified in two fragments. These fragments were assembled into a contiguous DAQ BGC and inserted into a receiver vector to create the vector pJB038EL in vitro using the NEBuilder HiFi DNA Assembly Cloning Kit (New England Biolabs). The plasmid pJB038EL was transferred into *S. coelicolor* M1152 through triparental conjugation. Based on HPLC and LC–MS analyses of the extract of the pJB038EL-harboring *S. coelicolor* M1152, production of DAQs **8**, **9**, **12**, and **13** (Figure 4A, bottom row) was confirmed. Although the yield was similar to that of the original host, the culturing efficiency was improved significantly to allow a simpler scale-up of the experiment. Moreover, the heterologous biosynthetic system provides the basis for generating structural analogs to solve the low solubility problem [78].

*Escherichia coli* is frequently used as a heterologous host of choice due to its fast growth, abundant genetic tools, and the deeper understanding we have of its metabolism. Almost all types of natural products have been attempted to be produced in *E. coli*, including macrolides, cyclic peptides, terpenes, and alkaloids [81]. Ikarugamycin **14** is an anticancer antibiotic [82]. A three-gene cassette (ikaABC) from the marine-derived *Streptomyces* sp. ZJ306 was identified to form the biosynthetic gene cluster that is responsible for the biosynthesis of **14** [80]. In order to prove the function of this gene cluster, Antosch et al. used *E. coli* for heterologous expression [79]. The gene cassette was incorporated into a suitable expression vector by homologous recombination, as shown in Figure 4B. Subsequently, the *ika* gene cluster was successfully expressed in *E. coli BAP1*, a strain developed for efficient phosphopantetheinylation of heterologous PKSs and NRPSs to generate holoenzymes [83].

Heterologous biosynthesis can activate silent metabolic pathways to facilitate the discovery of NPs that have not been identified before. It can also help increase the yield of NPs. With continued discovery of BGCs from marine microorganisms, which are often difficult to culture, research on heterologous host vectors will become crucial in promoting the greater application of BGCs to heterologous biosynthetic methods.

With the development of metabolic engineering, the use of various suitable microbial host systems to adapt the needs of specific designs of the target NP biosynthesis has achieved great success in enabling the identification and production of various NPs. However, due to the complexities of BGCs, a single engineered microbe typically cannot meet adequately the requirements of different biosynthetic pathways. Modular co-culture engineering is a newly developed method for natural product biosynthesis, which involves a complete modularization of the biosynthetic pathway. Every module is regulated in a different host to allow the modular expression of related genes to flexibly meet the different needs of different biosynthetic pathways [84,85]. This method makes full use of the production potential of various strains and decreases metabolic pressure on the strains and interference among the biosynthetic enzymes from different pathways. The modules can be taken from different pathways and adapted to the biosynthesis of different compounds. The system can also be adjusted to meet the requirements of heterologous and endogenous enzymes for the efficient synthesis of complicated natural products [86]. Modular co-culture engineering has been widely applied in the biosynthesis of natural products from plants and fungi [87,88]. Thus, it is anticipated that the approach can also provide a feasible scheme for the heterologous production of MNPs.

## 6. Combinatorial Biosynthesis

Under typical laboratory culturing conditions, many of the secondary metabolite biosynthetic pathways remain silent. Therefore, it is necessary to activate silent gene clusters and pathways to achieve biosynthesis of the corresponding NPs. Although heterologous expression can solve some problems, it is not immune to a number of other challenges, such as inadequate regulation of gene expression, difficulty in recruiting necessary modules and pathways, and inefficient transcription and translation of the biosynthetic genes. As a result, it often becomes necessary to engineer BGCs to improve their heterologous expression in order to synthesize the target natural products or their analogs sufficiently. Combined with fermentation techniques, heterologous expression could allow industrial production of NPs with medicinal values. To advance the effort further, we can also engage in combinatorial biosynthesis [89], which is an approach to engineer NP biosynthesis where the biosynthetic pathway encoded by a BGC of interest is engineered based on the inherent substrate promiscuity of the enzymes involved to produce NP analogs. Combinatorial biosynthesis can involve several approaches, comprising the metabolic engineering of microbes, the shuffling of enzyme-coding genes, and the feeding of small molecules to be used as non-native building blocks of the products being biosynthesized in order to generate a large library of products with novel chemical structures. In its simplest form, combinatorial biosynthesis can be achieved by engineering biosynthetic enzymes, particularly those that assume modular organization, such as PKSs and NRPSs. Since specific functions of biosynthetic enzymes can be mapped to the enzyme-coding genes or gene segments, these genes and gene segments can be shuffled or otherwise manipulated to generate engineered enzymes that can produce NP analogs. Genome mining of marine bacteria revealed that terpene (24%) and RiPP (21%) BGCs were amongst the most common BGCs in marine bacteria followed by PKS (14%), NRPS (13%), and β-lactone-related (7%) and global osmolyte ectoine (5%) BGCs [6]. As more and more MNP BGCs are cloned and the biosynthetic mechanisms are elucidated, it will become possible to incorporate these unique marine genetic resources into gene pools for combinatorial biosynthesis. Such an expansion of the scope of application of combinatorial biosynthesis would expedite the development of a wider variety of compounds with potential medicinal values. At present, a variety of combinatorial biosynthetic methods have been applied in the development of a library of NP analogs. Those methods can be categorized into (1) precursor-directed biosynthesis, (2) mutasynthesis, and (3) activation of silent pathways with suitable promoters (Figure 5). Despite a late start, combinatorial biosynthesis has attracted much attention in the MNP field and scored some important successes.

### 6.1. Precursor-Directed Biosynthesis

Exploiting the inherently relaxed substrate tolerance of some NP biosynthetic enzymes, it is possible to provide substrate analogs to a biosynthetic pathway to compete with the natural building blocks to give rise to NP analogs. This method is called precursor-directed biosynthesis [90]. This method introduces unnatural structural units into NPs but may synthesize both the original products as well as their analogs if the supply of the natural starting blocks cannot be blocked. It may require purification of the target product by chromatographic and other appropriate methods. For example, an unnatural analog of indolocarbazole (ICZ) has been successfully produced by this method. ICZs are a group of compounds with an indolo[2,3-a]pyrrolo[3,4-c]carbazole skeleton produced by nature. This class of NPs have been found in many species, including fungi, invertebrates, and terrestrial and marine actinomyces, and have strong physiological activities and interesting chemical structures [91]. One new ICZ, 3-hydroxy-K252d **17** (Figure 6), was obtained by feeding a culture of the marine *Streptomyces* strain OUCMDZ-3118 with a precursor 5-hydroxy-L-tryptophan, along with previously reported compounds 3-hydroxyholyrine A **15** and 3′-*N*-acetyl-3-hydroxyholyrine A **16** [92]. The structure of **17** was confirmed by 2D NMR correlations. The identity of the sugar moiety was determined to be l-rhamnose by gas chromatography–mass spectrometry (GC–MS) analysis of the acid hydrolysate of **17**. Cytotoxicity tests showed that **17** had moderate toxicity against the human lung cancer cell line A549 and the human breast cancer cell line MCF-7, with IC_50_ values of 1.2 μM and 1.6 μM, respectively [92].

Compared to conventional total chemical synthesis, precursor-directed biosynthesis permits more effective modifications of NPs, such as glycosylation, halogenation, and methylation. The method also allows more straightforward preparation of enantiomeric compounds due to the stereoselective nature of biological catalysts [93]. Precursor-directed biosynthesis has so far been shown to be a fast and excellent method to obtain natural product analogs.

### 6.2. Mutasynthesis

Although precursor-directed biosynthesis has been widely used, it can give rise to both original and modified products within the same producer, which makes the purification of the desired product out of the crude culture extract inconvenient. In mutasynthesis, certain biosynthetic enzymes are removed or modified so that the original NPs cannot be produced. Instead, only the desired analogs are produced by supplying specific substrate analogues to the system [94]. Mutasynthesis is conceptually similar to precursor-directed biosynthesis. The distinction is that production of the desired product is optimized in one of the following ways: (1) the enzymes are modified by protein engineering so that their substrate tolerance is adjusted to be able to accept analogs of the original substrates; (2) the genes involved in the formation of the original substrates are silenced to allow only the altered substrates to enter the desired biosynthetic pathway. The first approach will still require separation and purification of the altered product by chromatography or other methods. On the other hand, the second approach would abolish the formation of the original product. Hence, the crude extract would be highly enriched for the altered NP analogs. The drawback of the second approach is that it would require substantially more efforts to prepare the required engineered strain. Apart from product analog production, mutasynthesis can also be used to investigate the functions and properties of the enzymes involved in the biosynthesis of NPs of interest. The example of a successful application of mutasynthesis is the production of a series of analogs of an NP named A201A. A201A **18** is a nucleoside antibiotic with potent antibacterial activity that is produced by the deep-sea strain *Marinactinospora thermotolerans* SCSIO 00652 [95,96]. The biosynthetic gene cluster of **18** was identified in 2012, and it was discovered that MtdV is responsible for converting chorismic acid to 4-hydroxybenzoic acid (4HB) as a precursor for the biosynthesis of **18**. [97]. In order to verify the function of MtdV and test whether 4HB analogues can be used as precursors for the synthesis of product analogs in the absence of MtdV, this gene was inactivated and the analogs of 4HB were added to the culture of the Δ*mtdV* strain. HPLC analyses of the fermentation extracts revealed the presence of compounds **19**–**25** (Figure 7) that exhibited similar antibiotic activities to **18** (Table 1) [98]. Interestingly, no analogs were detected when 3-Br-4HB and 3-I-4HB were introduced, possibly because the large halogen atoms prevented those alternative building blocks from being accepted by the downstream enzymes to complete the formation of the analogs of **18**.

### 6.3. Activation of Silent Pathways Using Enhanced Promoters

Heterologous expression allows the expression of silent BGCs or BGCs from microbes that cannot be cultured. However, even with validated heterologous expression systems, some BGCs may still remain silent [99]. Often some extra genes are needed to improve or activate the expression of a BGC. Promoters are important factors for gene expression. In some cases, adding a strong promoter can stimulate the activation of BGC [100,101]. Below, we will discuss two examples of the successful application of promoter engineering to activate silent BGCs.

Through genome mining of the marine actinomycete *Streptomyces* sp. CNB-091, 38 kinds of BGCs that code for diverse biosynthetic pathways were found [102]. In an attempt to explore the biosynthetic potential and characterize the functions of those BGCs, Bauman and his colleagues (2019) focused on the s*pz* cluster, which was predicted to be responsible for the biosynthesis of streptophenazine-type compounds. The s*pz* BGC was transferred to a heterologous host *S. coelicolor* M1146 using the transformation-associated recombination (TAR) cloning method [103]. However, no production of novel secondary metabolites was detected in the initial expression test. The *spz* cluster was successfully reconstituted in *S. coelicolor* M1146 only after the BGC was refactored by introducing three promoter cassettes into the cluster. First, an engineered bidirectional promoter that combined strong synthetic promoters, p21 [104] and SP44 [105], to control the phenazine core biosynthetic genes was introduced to the cluster. The cluster was also modified with an *actI* [106] and an *ermE** [107] promoter to control the expression of the PKS genes and cytochrome bd oxidase genes, respectively. As a result, at least 38 streptophenazines were detected in the culture extract by LC–MS [103]. Similarly, Liu et al. introduced a strong promoter pSET152-*KasO* [108] into the silent combamide BGC *cbm* in *Streptomyces* sp. S10 and transferred the cluster to *Streptomyces* sp. LZ35 was modified by removing seven native PKS gene clusters and one native NRPS gene cluster to give a clean “background”. Through those efforts, they were able to obtain five combamide homologs A–E **26**–**30** (Figure 8) [109]. Promoters are essential for transcription, but in some cases native promoters present in a silent BCG may not be recognized by the heterologous host for transcription of the genes. Therefore, introducing suitable promoters into a BGC can help with necessary gene expressions and MNP biosynthesis.

For decades, different types of promoters have been used to modify gene expression. This method has been widely applied to regulate the expression of foreign genes. Use of inducible promoters allows the expression of target genes to be fine-tuned to improve protein production efficiency in cultured cells by adjusting the concentration of inducer chemicals. By optimizing heterologous protein production, these promoters can decrease the metabolic burden to prevent growth inhibition and death of the host microbe, thereby improving the overall yield of target metabolite production [101,110,111]. Different hosts favor different promoters. Selecting suitable promoters may yield twice the result with half the effort [112]. Accompanied with the ongoing development of bioinformatics tools and the deepening of our understanding of gene expression regulation mechanisms, machine learning is expected to generate more synthetic de novo promoters with higher activities and finer control [113].

Total synthesis is a relatively classical method for NP preparation. However, this method is not environmentally friendly, consumes substantial resources and time, and suffers from low overall yields. Furthermore, unmodified NPs may not be suitable for human applications in their original forms and may need to be modified to improve their pharmacokinetic and pharmacological properties. As the structures and functions of more and more BGCs become known, combinatorial biosynthesis will emerge as a viable alternative method to organic synthesis in introducing desired structural modifications to the NP scaffolds in a highly stereospecific manner to produce “improved” NPs. While combinatorial biosynthesis has promoted the fast development of MNPs, some problems remain. For example, engineered biosynthetic pathways need enzymes with high substrate tolerance. However, many enzymes have stringent substrate specificity. Likewise, engineered host cells may produce unexpected products or no product at all. Furthermore, the efficiency in reconstructing a biosynthetic system is rather low, especially when substrates with different structural elements are involved [89,114]. While these limitations still exist, combinatorial biosynthesis continues to be a powerful technique in engineering anabolic pathways to produce a wide range of NP derivatives which can be very useful for drug discovery and development.

## 7. Concluding Remarks

Extreme environmental conditions and complex habitats give rise to diversified secondary metabolites with unique structures and bioactivities. The ocean can provide numerous novel compounds, including a huge number of drug leads, pharmaceuticals, nutraceuticals, cosmetics, and agrochemicals [115]. Based on the conservative estimate by the investigators involved in the International Census of Marine Microbes (ICoMM), there are at least 20 million types of microbes in the sea, with an abundance of up to 10^30^ [116]. The number of marine microbes that have been identified and studied is very limited, and the number that has been used in pharmaceutical applications is even less. All over the world, there are active investigations of marine biodiversity and the development of MNPs.

Decades have passed since the initial activity screen to reach the point of NP discovery based on synthetic biological techniques. Although marine microbes are difficult to culture in the laboratory, we have seen the rapid accumulation of information on their genome sequences and in-depth analyses of the sequence data. Under such circumstances, transferring MNP BGCs into more conventional heterologous hosts is an important method in circumventing the culturability problem of marine microbes. *E. coli* and *Streptomyces* are the most commonly used heterologous hosts. However, a recently discovered novel microbe, *Vibrio natriegens*, was found to grow very rapidly, and is beginning to replace *E. coli* in laboratories as a workhorse microbe [19,117,118]. Various synthetic biology tools have been developed for preforming genetic engineering of *V. natriegens* [118,119,120], which provides a new alternative for the choice of heterologous hosts. The heterologous expression of NPs might be influenced by the specific regulation mechanisms present in a host [121]. Therefore, the exploration of new hosts and the development of gene- and genome-manipulation methods for such hosts are necessary to make full use of the potential of different BGCs.

However, the gene silencing that is common in native marine microbes and the low expression efficiency of BGCs from those organisms in heterologous hosts are still limiting factors that prevent the successful expression of BGCs for large-scale production of underexplored MNPs. One approach for solving such obstacles is to apply various techniques of synthetic biology to refactor the target biosynthetic pathway. Nowadays, there are various combinatorial biosynthetic methods that can be used for the development of MNPs. However, many of those methods, such as domain swapping, directed evolution, and pathway-level recombination, are still not so widely used [89]. Overall, heterologous expression and synthetic biology have provided new pathways for us to understand interesting biological mechanisms of chemical syntheses and discover many new NPs that would have been undetectable otherwise.

The slow development of NPs from marine microorganisms is in part due to limited sample availability but also due to the inadequacy of the research methods currently available. Nevertheless, there will be more new strains and NPs to be discovered with the advances in culturing techniques for marine microorganisms and microbial fermentation technologies [122]. Furthermore, breakthroughs in bioinformatics, metabolomics, and genetic engineering methods, as well as the continued discovery of MNP BGCs and subsequent in-depth investigations into the mechanisms of NP biosynthesis and the related enzymes that catalyze the complex sequences of reactions will advance our abilities to engineer and re-engineer complex NPs. With relentless cross-disciplinary investigations involving traditional genetic microbiological studies and the latest molecular, synthetic, and systems biological techniques, along with physicochemical and computational tools, MNPs are on the way to becoming resources of even greater importance in the prevention and treatment of major human diseases.

## Figures and Tables

**Figure 1 microorganisms-09-02551-f001:**
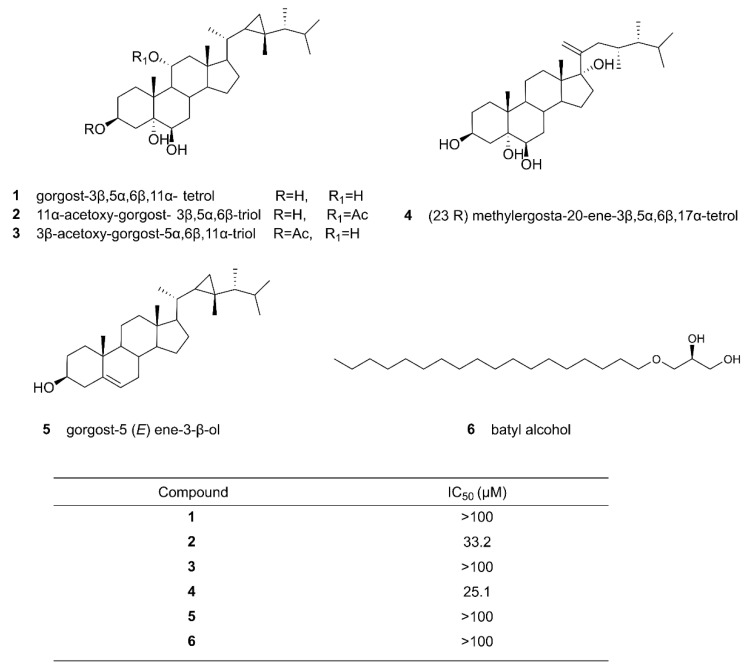
Chemical structures of the compounds isolated from Egyptian soft coral, *Heteroxenia fuscescens*, and their IC_50_ values against the MCF-7 (human breast adenocarcinoma) cell line [25]. Some of the compounds, such as **2** and **4**, were found to have moderate cytotoxic activities. Reproduced with permission from [25]. Copyright © 2019, Taylor & Francis, Leiden, The Netherlands.

**Figure 2 microorganisms-09-02551-f002:**
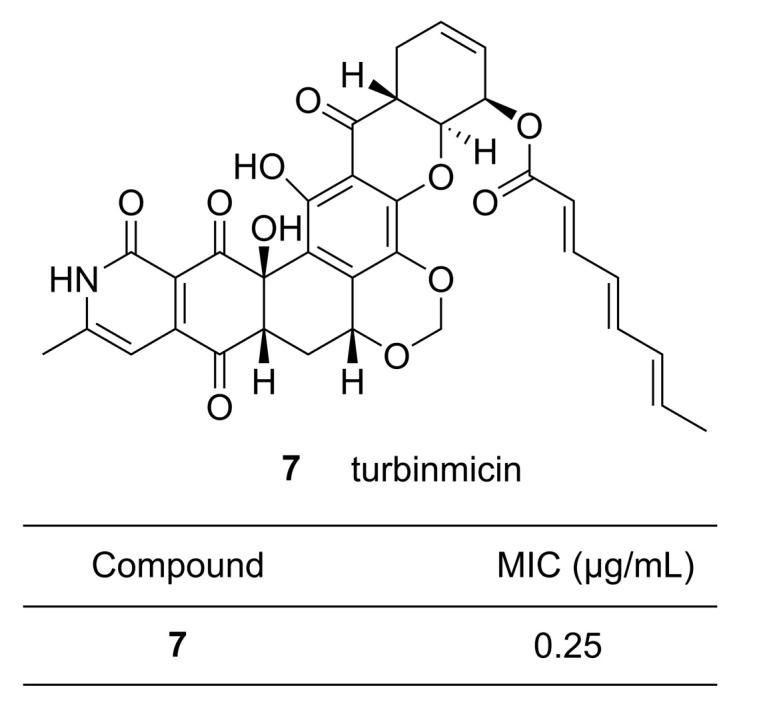
Chemical structure of a potential antifungal agent turbinmicin **7** that was obtained from the marine bacterium *Micromonospora* sp. isolated from the sea squirt *Ecteinascidia turbinata* with its minimum inhibitory concentration (MIC) against the panresistant *Candida auris* strain B11211 [55]. Reproduced with permission from [55]. Copyright © 2020, The American Association for the Advancement of Science, Washington, DC, USA.

**Figure 3 microorganisms-09-02551-f003:**
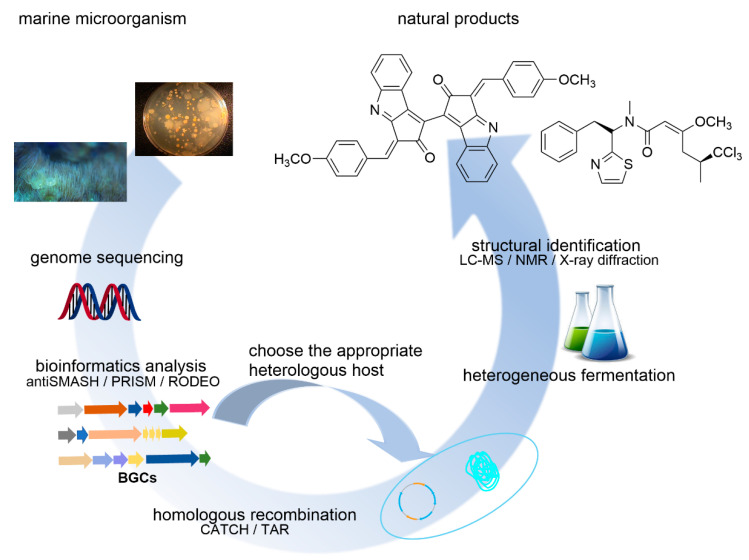
Schematic diagram showing the general strategy taken for heterologous expression of biosynthetic gene clusters (BGCs) that are identified through genome or environmental DNA sequence analyses using various available tools, such as antiSMASH [42], PRISM [43], and RODEO [44]. Typically, the target BGC is amplified from the genome and cloned into a suitable vector, where the BGC is placed under the control of a promoter that facilitates the expression of the genes in the cluster. Techniques such as TAR (transformation-associated recombination) [64] and CATCH (Cas9-assisted targeting of chromosome segments) [65,66] can be used to expedite this step. Once the host successfully expresses the BGC, the culture is examined by liquid chromatography–mass spectrometry (LC–MS) and other analytical methods to determine whether the target secondary metabolites are produced by the heterologous host. Adapted with permission from ref. [71]. Copyright © 2021 Year ROYAL SOCIETY OF CHEMISTRY, London, UK.

**Figure 4 microorganisms-09-02551-f004:**
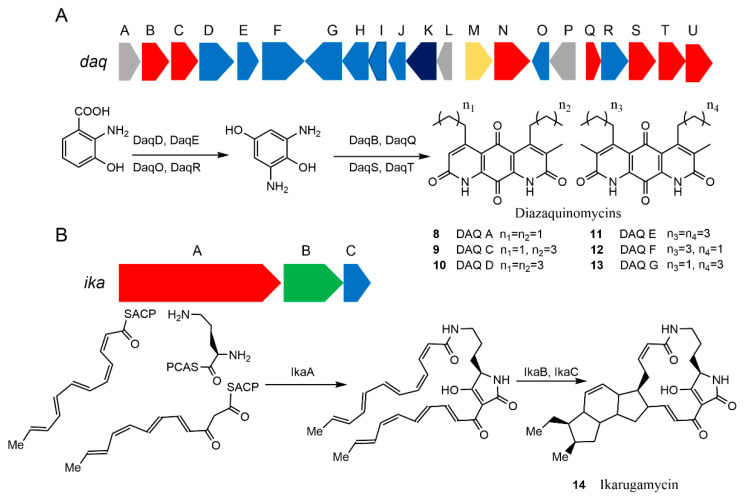
Examples of marine biosynthetic gene clusters (BGCs) that are successfully expressed in heterologous hosts. (**A**) The BGC and the simplified pathway for the biosynthesis of the diazaquinomycin (DAQ) family of compounds A and C–G [76]. (**B**) The BGC and the pathway for the biosynthesis of ikarugamycin [79,80]. Two molecules of hexaketides are generated by the iterative polyketide synthase portion of IkaA and are condensed with one molecule of ornithine by the condensation domain of the nonribosomal peptide synthetase segment of IkaA to form the intermediate that undergoes reductive cyclization reactions proposed to be catalyzed by IkaB and IkaC to yield ikarugamycin 14. Reproduced with permission from [76]. Copyright © 2019, American Chemical Society, Washington, DC, USA. Reproduced with permission from [79]. Copyright © 2014 WILEY-VCH Verlag GmbH & Co. KGaA, Weinheim, Germany.

**Figure 5 microorganisms-09-02551-f005:**
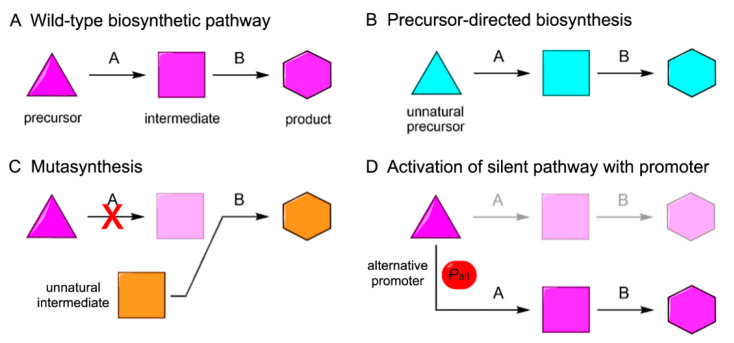
Schematic diagram showing different strategies employed for combinatorial biosynthesis of a secondary metabolite for generation of its analogs. A and B are biosynthetic enzymes. The triangle, square, and hexagon represent the precursor, intermediate, and product, respectively. Magenta indicates wild-type metabolites, while cyan and orange represent altered metabolites generated by different methods of combinatorial biosynthesis. Semitransparent parts represent inactive steps or absent metabolites. (**A**) Wild-type biosynthetic pathway. (**B**) Biosynthetic pathway altered by precursor-directed biosynthesis, where an alternative precursor molecule to initiate the biosynthetic steps is introduced to form a modified product. (**C**) Biosynthetic pathway altered by mutasynthesis, where incorporation of the wild-type precursor is blocked by inactivating A and an alternative pathway intermediate is provided to generate an altered product. (**D**) Biosynthetic pathway altered by activation of silent pathway by promoter engineering. An alternative promoter (*P*_alt_) can be introduced to a silent gene cluster to activate the biosynthesis of the product.

**Figure 6 microorganisms-09-02551-f006:**
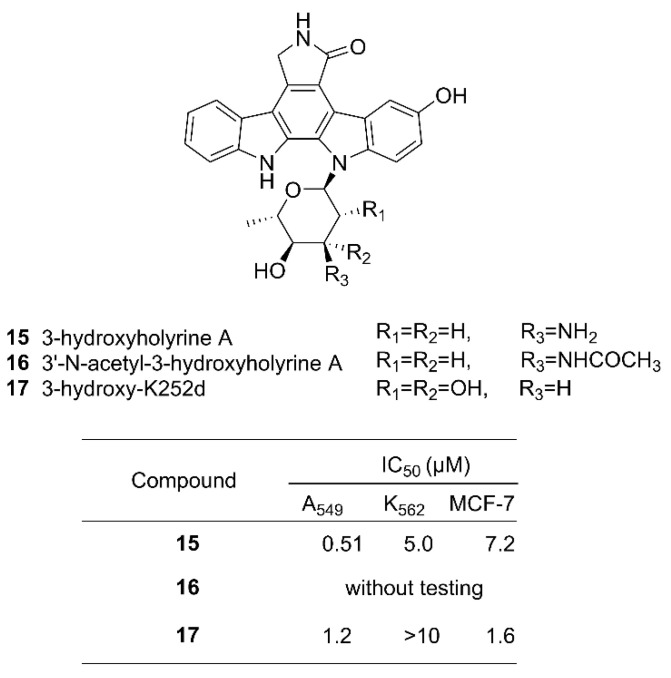
Chemical structures of hydroxy-substituted indolocarbazoles **15**–**17** obtained by feeding a precursor 5-hydroxy-L-tryptophan to the marine *Streptomyces* strain OUCMDZ-3118. The IC_50_ values of these products against the A549 (human lung adenocarcinoma), K562 (human erythroleukemia), and MCF-7 cell lines are also given [92]. Reproduced with permission from [92]. Copyright © 2018, MDPI, Basel, Switzerland.

**Figure 7 microorganisms-09-02551-f007:**
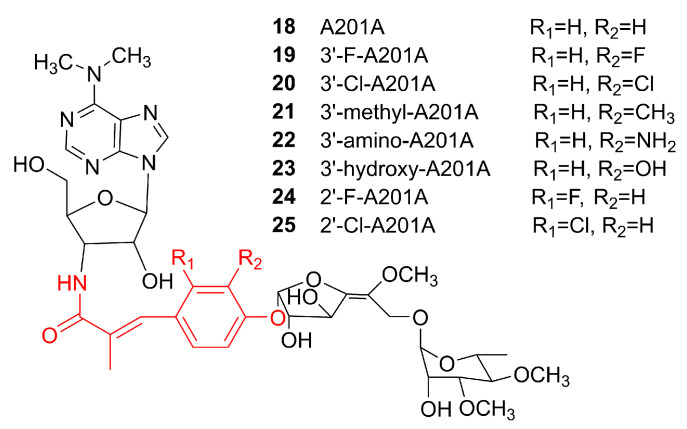
Chemical structures of A201A **18** and its analogs **19**–**25** obtained from the deep-sea derived *Marinactinospora thermotolerans* SCSIO 00652 by a mutasynthetic approach. Various 4-hydroxybenzoic acid analogs were fed to the strain with a deletion of the key chorismite lyase gene *mtdV* [98]. Copied with permission from [98]. Copyright © 2020, Royal Society of Chemistry.

**Figure 8 microorganisms-09-02551-f008:**
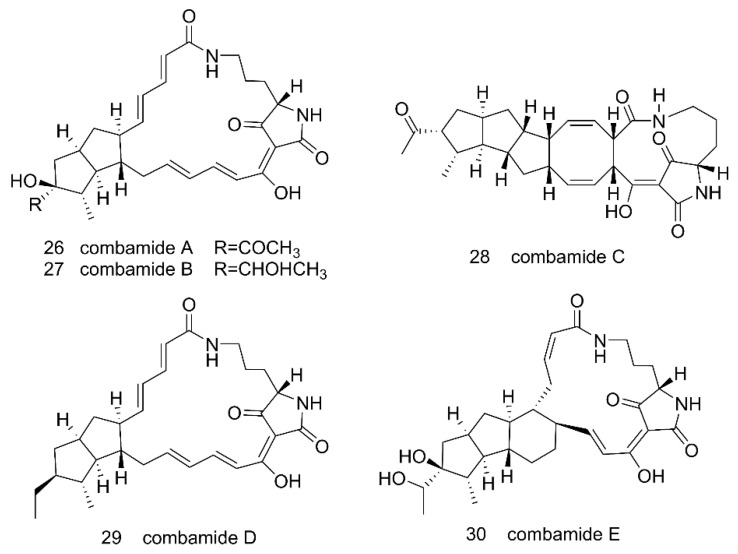
Chemical structures of combamide A–E **26**–**30** obtained by heterologously expressing the *cbm* biosynthetic gene cluster from *Streptomyces* sp. S10 in an engineered *Streptomyces* sp. LZ35 strain [109]. Unfortunately, most of those compounds were not very active against human cell lines or bacterial strains [109]. Reproduced with permission from [109]. Copyright © 2018, American Chemical Society.

**Table 1 microorganisms-09-02551-t001:** The minimum effective concentration values (μg/mL) of compounds **18**–**20** against various Gram-negative and positive bacterial strains [98]. Copied with permission from [98]. Copyright © 2020, Royal Society of Chemistry.

Strain	Compound and Its MIC (µg/mL)		
	18	19	20
*E. coli* ATCC25922	>128	>128	>128
*Staphylococcus aureus* ATCC 29213	2	1	1
*Micrococcus luteus*	8	2	4
*Bacillus subtilis* BS01	16	16	16
*Bacillus thuringiensis* BT01	2	4	8

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
