# Peer review of "Advances in Biosynthesis of Natural Products from Marine Microorganisms"

_microorganisms, 2021, doi:10.3390/microorganisms9122551_

Round 1

Reviewer 1 Report

The manuscript submitted by Zhou et al. focused on the recent advances in marine microbial natural products (MNPs) biosynthesis. Authors referred to different challenges during traditional MNPs discovery and new approaches to tackle these problems such as meta/genome mining of marine microorganisms using bioinformatics tools to detect the new natural product biosynthesis gene clusters (BGCs). Next, they explained the recent heterologous and combinatorial biosynthesis approaches for discovery of novel MNPs.

The manuscript is clear and includes most important approaches used recently in the field. It is also of interest for simple step-by-step explanation of different approaches used in the field. However, authors missed to compare their review with most recent reviews on marine natural product discovery. There are several reviews focused on recent biosynthetic studies and methodologies of marine natural products. Therefore, I highly recommend the authors to distinguish their review from other recent reviews on this field.

Below, I have some comments for authors which needs to be addressed for better readability and correctness:

Line 37: Natural products discovered from marine environments are not only from deep-sea and mostly isolated from samples taken in shallow waters. The extreme environments therefore can also be considered in high or low temperatures, high-pressure, low pH, high salt concentrations, and etc.

Line 91: Please write the name of the structures when used for the first time.

Line 95: Please use the same style of the structures in different figures. Figure 1 has smaller size of the structures and these structures can be placed in better way to see almost the same space in between.

Line 132: Other recent tools or platforms can be added here, such as ARTS and Paired Omics Data Platform

https://doi.org/10.1093/nar/gkaa374

https://doi.org/10.1038/s41589-020-00724-z

Line 196: Please give the long form of PTN pathway here.

Line 200: Again, please use the same style for the figures and the structures. Figure 2 has structures bigger than other figures. The table below the structure is also different. Please use same font, size, and etc. for the structures.

Line 208: rediscovery is a problem in general and does not interrelate with only marine microorganisms.

Line 217: …”gene sequencing” and chemical structure identification should complement each other. Here the term ”gene sequencing” is not correct. Instead, “prediction of BGCs based on genome mining approach” can be used.

Line 218: Again, “gene sequencing” is not correct. Instead, Detection of BGCs can predict ….

Line 256: Please use structures in similar styles as in other figures.

Line 274: MAUVE is not a tool for comparing the gene clusters. It can be used to map a genome against a reference genome. There is a tool called MultiGeneBlast for identification of homologs of multigene modules such as operons and gene clusters.

Line 346: Please give a reference for this sentence, otherwise, genome mining of the marine bacteria revealed that BGCs related to terpene (24%) and RiPPs (21%) biosynthesis were amongst the most common BGCs within marine bacteria followed by PKS, NRPS and betalactone-related pathways, as well as the biosynthesis of the global osmolyte ectoine.

https://doi.org/10.1039/D0OB01677B

Line 395: Again, please use the same style of the structure and the text used for annotations.

Line 412: “In this way, mutasynthesis can be applied to obtain relatively pure natural product analogs.” This can not be true in terms of obtaining pure compounds or analogs. Probably, authors meant solely producing the altered analogs.

Line 439: Better to use genome mining study instead of sequencing study.

Line 459: Same style please! If you write the number and the name of the structure, do it in all figures.

Line 479: pharmacological activity not pharmaceutical activity.

Author Response

Reviewer #1

General Comment

  1. Authors missed to compare their review with most recent reviews on marine natural product discovery. There are several reviews focused on recent biosynthetic studies and methodologies of marine natural products. Therefore, I highly recommend the authors to distinguish their review from other recent reviews on this field.

       We thank the reviewer for the comment. To address the reviewer's suggestion, we have included other recent reviews in our citations and concisely highlighted the features that distinguish our review from others (at the end of section 1). The description is as follows:

Compared with previous reviews [6,16], our review covers a broader range of marine microorganisms and focuses on practical strategies and technologies for solving the problems that may hamper the researches of MNPs.

Specific Comments

  1. Line 37: Natural products discovered from marine environments are not only from deep-sea and mostly isolated from samples taken in shallow waters. The extreme environments therefore can also be considered in high or low temperatures, high-pressure, low pH, high salt concentrations, and etc.

       Thanks a lot for the reviewer to point out this defect in our manuscript. We have changed the descriptions accordingly. The current description is as follows:

The extreme environment in the sea, characterized by various conditions, such as high or low temperatures, high pressure, low pH, and high salt concentrations [5]

  1. Line 91: Please write the name of the structures when used for the first time.

       Thanks a lot for the reviewer to point out this defect in our manuscript. We have provided the full names for compound 2 as 11α-acetoxy-gorgost-3β,5α,6β-triol and 4 as (23R) methylergosta-20-ene-3β,5α,6β,17α-tetrol.

  1. Line 95: Please use the same style of the structures in different figures. Figure 1 has smaller size of the structures and these structures can be placed in better way to see almost the same space in between.

       Thanks a lot for the reviewer to point out this problem in our manuscript. We have re-designed all the figures and tables to have the same style and size.

  1. Line 132: Other recent tools or platforms can be added here, such as ARTS and Paired Omics Data Platform.

       https://doi.org/10.1093/nar/gkaa374    https://doi.org/10.1038/s41589-020-00724-z

       Thanks a lot for this suggestion. We have mentioned the said tools in the revised manuscript. The description is as follows:

Paired Omics Data Platform (PoDP, https://pairedomicsdata.bioinformatics.nl, accessed on 18 November 2021) is a new tool for paired genomic and metabolomic data mining and for establishing large-scale genome–metabolome associations that facilitates elucidation of the structures of metabolites and the biosynthetic genes encoded on microbial genomes [40]. Specific for antibiotics genome mining, Antibiotic Resistant Target Seeker (ARTS, http://arts.ziemertlab.com, accessed on 18 November 2021) has been developed and accelerated the discovery of new antibiotics [41]. ARTS specializes in detecting and analyzing resistance factors associated with BGCs that can provide insight into whether or not the BGCs encode antibiotics.

  1. Line 196: Please give the long form of PTN pathway here.

       Thanks a lot for this suggestion. We have provided the full name of the PTN pathway as the pentostatin pathway.

  1. Line 200: Again, please use the same style for the figures and the structures. Figure 2 has structures bigger than other figures. The table below the structure is also different. Please use same font, size, and etc. for the structures.

       We have re-designed all the figures and tables to have the same style and size.

  1. Line 208: rediscovery is a problem in general and does not interrelate with only marine microorganisms.

       Thanks a lot for the reviewer to raise this point. We have rewritten the section.(p. 6). The description is as follows:

Moreover, rediscovery of known compounds is a common problem in NP discovery [52].

  1. Line 217: …”gene sequencing” and chemical structure identification should complement each other. Here the term ”gene sequencing” is not correct. Instead, “prediction of BGCs based on genome mining approach” can be used.

       Thanks a lot for pointing this out. We have revised the sentence accordingly. (p. 7). The description is as follows:

prediction of BGCs based on genome mining approach and chemical structure identification should complement each other.

  1. Line 218: Again, “gene sequencing” is not correct. Instead, Detection of BGCs can predict …..

       Thanks a lot for pointing out this inadequate term usage in our manuscript. We have revised and expanded the line.(p. 7).The description is as follows:

Identification of BGCs can supply a large number of new structures

  1. Line 256: Please use structures in similar styles as in other figures.

       We have re-designed all the figures and tables to have the same style and size.

  1. Line 274: MAUVE is not a tool for comparing the gene clusters. It can be used to map a genome against a reference genome. There is a tool called MultiGeneBlast for identification of homologs of multigene modules such as operons and gene clusters.

       We agree with the reviewer that our description was not very accurate. We have revised the  passage to reflect the points raised by the reviewer. (p. 10). The description is as  follows:

Here the MAUVE alignment [71] was applied to the genome sequence of the marineStreptomyces sp. strain F001 against that of the freshwaterMicromonospora sp. strain B006 to find a shared gene cluster between the two DAQ-producing strains.

  1. Line 346: Please give a reference for this sentence, otherwise, genome mining of the marine bacteria revealed that BGCs related to terpene (24%) and RiPPs (21%) biosynthesis were amongst the most common BGCs within marine bacteria followed by PKS, NRPS and betalactone-related pathways, as well as the biosynthesis of the global osmolyte ectoine.

       https://doi.org/10.1039/D0OB01677B

       Thanks a lot for the reviewer to point out this oversight in our manuscript. We have added the reference to this sentence.(p. 13). The description is as follows:

Genome mining of marine bacteria revealed that terpene (24%) and RiPPs (21%) BGCs were amongst the most common BGCs in marine bacteria, followed by PKS (14%), NRPS (13%), b-lactone-related pathways (7%), and the global osmolyte ectoine (5%) BGCs [6].

  1. Line 395: Again, please use the same style of the structure and the text used for annotations.

       We have re-designed all the figures and tables to have the same style and size.

  1. Line 412: “In this way, mutasynthesis can be applied to obtain relatively pure natural product analogs.” This can not be true in terms of obtaining pure compounds or analogs. Probably, authors meant solely producing the altered analogs.

       Thanks a lot for the reviewer for this suggestion. We have revised the sentence accordingly. (p. 16). The description is as follows:

On the other hand, the second approach would abolish the formation of the original product. Hence the crude extract would be highly enriched for the altered NP analogs

  1. Line 439: Better to use genome mining study instead of sequencing study.

       We agree that this description was not accurate and have changed accordingly. (p. 18). The description is as follows:

Through genome mining of the marine actinomyceteStreptomyces sp. CNB-091

  1. Line 459: Same style please! If you write the number and the name of the structure, do it in all figures.

       We have re-designed all the figures and tables to have the same style and size.

  1. Line 479: pharmacological activity not pharmaceutical activity.

       Thanks a lot for the reviewer to point this mistake. We have corrected it accordingly.

Reviewer 2 Report

The authors describe in this review the recent developments in marine natural product research, with emphasis in compound discovery and the new bioinformatic approach to improve the quantity and quality of microbial products. The idea is very interesting and open new point of view on the marine microbial products. However, some points merit to be more clarified or rewrite because give some confusion to the reader. Furthermore, some aspects is not treated by authors such as the microalgae compounds and ther bioproducts. Moreover, it not be clear why the author mentioned the extreme environments.

Line 35-36 Maybe the authors should be specify if intend microbial compounds.

Line 37 “The extreme environment,…..” it is refer to the deep marine environments? The sentence should be rewrite more clearly

Line 52-52 “The difficulty of reaching the marine environment…..” the authors should specify that marine environment is composed by costal, abyssal, shallow water… not all the marine environment is difficult to reach please take in mind this consideration.

Line 119 should you cited also the microalgae (Liang et al 2019 DOI: 10.1080/10408398.2018.1455030 Nicolo et al 2021 DOI: 10.1111/jam.15223)

Line 191 “which was 191 a microbiome constituent of the sea squirt Ecteinascidia turbinata” please rewrite

Line193 “concentration (MIC) of 7 against the panresistant C. auris strain B11211 was 0.25 µg/mL” “7??” is referred to…???

 Line 206 “Due to the special environment that marine microorganisms live in, such as pressure, 206 temperature, pH, salt, and high metal concentrations [43]” please rewrite

Line 423 “as compound 18” 18 is referred to…?

Line 426 please separate the Fig from the table, the table should be more readable.

Line 506 “streptomyces” please change to “Streptomyces”

All the tables should be formatted.

Author Response

Reviewer #2

General Comments

  1. Some points merit to be more clarified or rewrite because give some confusion to the reader.

       We hope incorporating the valuable suggestions from the reviewers and our own revisions have improved the clarity and understandability of our manuscript.

  1. Some aspects is not treated by authors such as the microalgae compounds and ther bioproducts.

       Thanks a lot for this suggestion. We have mentioned microalgae in the section “3. Genome-independent MNPs discovery”. (p. 3).We have also added a mention of microalgae in the section "5. Heterologous biosynthesis". (p. 7). The descriptions are as follows:

Similarly, de Vera et al. (2018) cultured 33 microalgae strains and prepared the extracts to screen for antimicrobial, anti-proliferative, and apoptotic potentials [23]. Through the study, they were able to identify several hit compounds for future drug development.

Recently, microalgae have been applied as heterologous hosts to produce proteins at high expression levels through sequence optimization [64]and using secretory signal peptides [65].

  1. It not be clear why the author mentioned the extreme environment.

       Thanks a lot for this question. We mentioned extreme environment, because unique ecological parameters associated with extreme environment drive the adaption of marine organisms, including their production of marine specific molecular compounds. We added and clarified explanations on this point in our revised manuscript (p. 1 and 7) and added References 6 and 16 in addition to 5.

Specific Comments

  1. Line 35-36 Maybe the authors should be specify if intend microbial compounds.

       Thanks a lot for this suggestion. We have changed the description accordingly. (p. 1). The description is as follows:

marine natural products (MNPs), especially microbial secondary metabolites, have attracted wide attention [4].

  1. Line 37 “The extreme environment,…..” it is refer to the deep marine environments? The sentence should be rewrite more clearly.

       Thanks a lot for the reviewer to point this out. We have revised it accordingly. (p. 1). The description is as follows:

The extreme environment in the sea, characterized by various conditions, such as high or low temperatures, high pressure, low pH, and high salt concentrations is thought to endow marine secondary metabolites with novel chemical structures and unique physiological properties because of the major driving forces for the evolution of regulated biosynthetic traits [6]

  1. Line 52-52 “The difficulty of reaching the marine environment…..” the authors should specify that marine environment is composed by costal, abyssal, shallow water… not all the marine environment is difficult to reach please take in mind this consideration.

       Thanks a lot for the reviewer to point out this oversight in our manuscript. We have revised the manuscript for improved clarity. (p. 2). The description is as follows:

Unlike terrestrial sample collection, the difficulty in reaching the marine environment, especially in the deep sea, poses problems for sample collection.

  1. Line 119 should you cited also the microalgae (Liang et al 2019 DOI: 10.1080/10408398.2018.1455030 Nicolo et al 2021 DOI: 10.1111/jam.15223).

       Thanks a lot for this advice. We have added the references.

  1. Line 191 “which was 191 a microbiome constituent of the sea squirt Ecteinascidia turbinata” please rewrite.

       Thanks a lot for this suggestion. We have revised the sentence accordingly. (p. 6). The description is as follows:

Based on titers, MS, and NMR analysis, they discovered a promising antifungal drug turbinmicin (Figure 2) from Micromonospora sp., a member of a sea squirt microbiome.

  1. Line193 “concentration (MIC) of 7 against the panresistant C. auris strain B11211 was 0.25 µg/mL” “7??” is referred to…???.

       refers to the compound turbinmicin as mentioned in the text and in Figure 2.

  1. Line 206 “Due to the special environment that marine microorganisms live in, such as pressure, 206 temperature, pH, salt, and high metal concentrations [43]” please rewrite.

       Thanks a lot for this suggestion. We have rewritten this sentence accordingly. (p. 7). The description is as follows:

Due to special conditions of the environment in which marine microorganisms typically inhabit, such as extreme temperature, pressure, pH, and salt concentration [5]

  1. Line 423 “as compound 18” 18 is referred to…?.

       18 refers to the compound A201A as mentioned in the text and in Figure 7.

  1. Line 426 please separate the Fig from the table, the table should be more readable.

       Thanks a lot for the reviewer for the suggestion. We have divided the original figure (Figure 7) into a figure and a table (Table 1).

  1. Line 506 “streptomyces” please change to “Streptomyces”.

       Thanks a lot for the reviewer to point out this mistake. We have changed it accordingly.

  1. All the tables should be formatted.

       Thanks a lot for pointing out this defect in our manuscript. We have standardized the formatting of all of the tables.

Round 2

Reviewer 2 Report

The authors increased the readability of the text and all questions were addressed. I consider this manuscript acceptable in its present form.

Author Response

Reviewer #2

The authors increased the readability of the text and all questions were addressed. I consider this manuscript acceptable in its present form.

Response:

We thank the reviewer for this encouragement.
